# Dopamine in Health and Disease: Much More Than a Neurotransmitter

**DOI:** 10.3390/biomedicines9020109

**Published:** 2021-01-22

**Authors:** Rafael Franco, Irene Reyes-Resina, Gemma Navarro

**Affiliations:** 1Neurodegenerative Diseases, CiberNed. Network Research Center, Spanish National Health Institute Carlos III, Valderrebollo 5, 28031 Madrid, Spain; g.navarro@ub.edu; 2Department of Biochemistry and Molecular Biomedicine, University of Barcelona, 08028 Barcelona, Spain; 3Department of Biochemistry and Physiology, Faculty of Pharmacy and Food Science, University of Barcelona, 08028 Barcelona, Spain

**Keywords:** microbiota, intestinal flora, Parkinson’s disease, immune system, drug development, receptor heteromers, G protein-coupled receptors, L-DOPA, inflammation, T-cell activation

## Abstract

Dopamine is derived from an amino acid, phenylalanine, which must be obtained through the diet. Dopamine, known primarily to be a neurotransmitter involved in almost any higher executive action, acts through five types of G-protein-coupled receptors. Dopamine has been studied extensively for its neuronal handling, synaptic actions, and in relation to Parkinson’s disease. However, dopamine receptors can be found extra-synaptically and, in addition, they are not only expressed in neurons, but in many types of mammalian cells, inside and outside the central nervous system (CNS). Recent studies show a dopamine link between the gut and the CNS; the mechanisms are unknown, but they probably require cells to act as mediators and the involvement of the immune system. In fact, dopamine receptors are expressed in almost any cell of the immune system where dopamine regulates various processes, such as antigen presentation, T-cell activation, and inflammation. This likely immune cell-mediated linkage opens up a new perspective for the use of dopamine-related drugs, i.e., agonist–antagonist–allosteric modulators of dopamine receptors, in a variety of diseases.

## 1. Introduction

Mammalian endogenous monoamines are very interesting compounds. They act as neurotransmitters and as regulatory molecules that participate in keeping homeostasis, but that may become part of the problem in some diseases. An example is adrenaline/noradrenaline, which are neurotransmitters of sympathetic neurons but which, in addition, are secreted by the adrenal gland to maintain homeostasis. In tumors of the adrenal gland, overproduction of these monoamines leads to many of the serious symptoms of the patients. Dopamine (DA) is more known as a neurotransmitter, although it also acts as a compound that helps in maintaining homeostasis. However, a difference with adrenaline/noradrenaline is that DA seems to act more in the paracrine mode than in an endocrine-type fashion.

Monoamines (adrenaline, noradrenaline, DA, etc.) were first detected in the brain. The first detection of a compound that was likely DA was in 1951 in the brain of different animals (humans included) [1]. Similar studies but addressing regional distribution failed to detect DA in areas of low noradrenaline content (e.g., the caudate putamen) [2]. A few years later, the compound, which was neither noradrenaline nor adrenaline, was isolated and identified as DA by paper chromatography [3,4]. At that moment, the main question was the physiological role of DA. A very important piece of information came from the discovery that L-DOPA, which is the precursor of DA (Figure 1), was facilitating waking up after hexobarbital anesthesia; the authors of the study postulated that DA was the active compound whose precursor was L-DOPA [5]. Research in the late 1950s was key in demonstrating that DA had specific functions in and out of the brain, and that L-DOPA was a precursor that upon administration to animals could be converted into DA [6,7].

A key discovery was the link between dopamine deficiency and Parkinson’s disease (PD). Several laboratories participated, some to discover that the cause of PD was in the brain structures related to the striatum, and others showed that it was a lack of dopamine in the substantia nigra which, in turn, led to the depletion of DA in the striatum. Soon after these early discoveries, L-DOPA was proposed as a medication to combat the symptoms of PD (see Ref. [8] for a detailed account of the discovery of DA and of DA deficiency in the Parkinsonian brain) [9,10]. Studies with animal models of PD have suggested that the motor disturbances are due to a disbalance in the so-called direct and indirect striatal pathways, in which neurons projecting to the globus pallidus and substantia nigra pars reticulata have two different types of dopamine receptors (D_1_ or D_2_; five DA receptor types have been discovered; see below). This view has been recently challenged by immunochemical-based assays in non-human primates showing that striatal neurons may express the two receptor types [11], and by single-axon tracing studies (also in non-human primates), indicating evidence against a dual striatofugal system [12].

Dopamine is now arising as one of the most relevant neurotransmitters, as it seemingly participates directly or indirectly in almost any physiological function occurring in the central nervous system (CNS). This review presents information about dopamine that has mainly been obtained from enzymatic studies, and about the link with PD, without forgetting the actions that DA exerts in the periphery. The review also highlights the cell-based communication that links the gut and the CNS.

## 2. Dopamine, Dopamine Receptors and Catechol-Related Enzymes

DA, or 4-(2-aminoethyl)-1,2-benzenediol, is one of the main neurotransmitters in the mammalian nervous system. In the central nervous system, it is produced and released by the so-called dopaminergic neurons, which are found in different brain areas but are especially abundant in the substantia nigra. Dopaminergic neurons of the substantia nigra pars compacta project to the striatum, where specific receptors are activated and the signal is transmitted by projection neurons and further circuitry to the pallidus, the thalamus, and the substantia nigra pars reticulata. DA participates in almost any centrally controlled event, from motor control to cognition.

Overall, DA acts via two well-described mechanisms, namely, wiring and volume transmission. Wiring transmission participates in releasing DA to the synaptic cleft, where it acts on postsynaptic DA receptors. Volume transmission occurs when extracellular DA arrives to neurons other than those postsynaptically located; i.e., by diffusion, DA reaches receptors in other neurons (or glial cells) that are not in direct contact with the cell that has released the neurotransmitter [13,14,15,16]. This means that the brain contains a certain degree of dopaminergic tone resulting from the DA that, released to the extracellular medium, activates extrasynaptic receptors in different cells across the brain.

So far, five DA receptors have been described (D_1_, D_2_, D_3_, D4 and D_5_), which belong to the superfamily of G protein-coupled receptors (GPCRs). For many years it was assumed that DA’s role in the brain consisted of producing variations in intraneuronal cAMP levels, and in the activation/inactivation of protein kinase, which in turn regulates the activity of a protein that is considered key in dopaminergic neurotransmission, DA- and cAMP-regulated phosphoprotein of 32 kDa molecular weight (DARPP32). The cognate protein of D_1_ and D_5_ receptors is Gs, whose engagement activates adenylate cyclase, thus leading to increases in cytosolic cAMP levels. Instead, the cognate protein of the D_2_, D_3_ and D_5_ receptors is Gi, whose engagement inactivates adenylate cyclase, thus leading to decreases in cytosolic cAMP levels [17]. Gs or Gi coupling may occur in cells other than neurons because DA receptors are found in many cell types (even in the periphery), thus substantiating the interpretation that DA is more than a neurotransmitter. As shown below, there are splice variants of dopamine receptors; some of them are “natural”, i.e., occurring in all individuals, and some vary from individual to individual and may be associated with impulsive behaviors or with the risk of addiction. Splice variants are mainly described for the D_2_ and the D_4_ receptors; importantly, other gene polymorphisms described for all five receptors, including single nucleotide polymorphism, may be associated with a variety of addictions (drugs, alcohol, etc.) and obesity [18,19,20,21,22,23,24,25]; other non-dopamine-receptor-related factors also influence addictive behaviors.

## 3. Features of DA Receptors That Are Important in Dopaminergic Transmission in Both Health and Disease

For decades it was assumed that the DA link to calcium-mediated actions was indirect. However, the laboratory of Susan George provided evidence that the D_1_–D_2_ receptor assemblies couple to Gq/11 instead of coupling to Gs or Gi [26,27]. Gq coupling allows calcium mobilization, this being a direct link between DA and actions mediated by the ion. The controversy as to whether D_1_–D_2_ receptor interactions do not occur in primates was solved by the demonstration that there are a significant percentage of neurons expressing D_1_–D_2_ receptor complexes in the *Macaca fascicularis* monkey model [11]. A physiological role of D_1_–D_2_ receptor heteromers was demonstrated in the healthy brain, and a pathophysiological one in models of addiction [11,28,29,30]. In addition, the same laboratory has reported that expression differences depending on the sex affect anxiety- and depression-like neuropsychiatric diseases [31].

The variety of effects in the healthy brain and the therapeutic potential of targeting DA receptors in diseases of the nervous system is sustained by the myriad possibilities derived from the occurrence of complexes formed by DA receptors, or by DA receptors and other cell surface receptors. DA’s action in a given cell will depend on the DA receptors expressed on the plasma membrane and on the expression of complexes in which these receptors participate. Indeed, each receptor–receptor heteromer conveys a different signaling when activated by DA, i.e., each receptor heteromer has its own physiological role and pharmacological properties [32,33,34]. An example that is relevant for Parkinson’s disease is the occurrence of striatal neurons expressing D_1_ and adenosine A_1_ receptors [35], or D_2_ and adenosine A_2A_ receptors [36]. The antagonism between the dopaminergic transmission and purinergic regulation of neurotransmitter release is, in part, due to the occurrence of these heteromers [37,38,39,40].

Other homotropic heteromers that have been so far reported include D_1_–D_3_ [41,42], D_2_–D_3_ [43], D_2_–D_5_ [28] and D_2_–D_4_ [44]. Apart from the A_1_–D_1_ and the A_2A_–D_2_, other heterotropic receptor heteromers include the D_1_–histamine H_3_ [45], the D_2_–histamine H_3_ [46], the D_4_–adrenergic [47], etc.; for a complete list of DA receptor-containing complexes, see http://www.gpcr-hetnet.com/ [48]. As an example of the physiological role of DA-containing heteromers, those involving some adrenergic receptors are involved in the circadian regulation of melatonin production by the pineal gland [48].

Despite the fact that there are few examples of isoforms in GPCRs, the mRNAs for the D_2_ and D_4_ receptors may suffer differential splicing, thus giving rise to various isoforms. The most studied is the D_2_ short and the D_2_, which differ in the size of the third intracellular loop [49]. Interestingly, the D_2_ short may be expressed presynaptically to regulate neurotransmitter release, thus acting as an autoreceptor [50,51]. The circa 20 variants of the D_4_ receptor are due to a hypervariable region of the D_4_ gene; they come from 27 haplotypes [52,53] and one of them, the D_4.7_, is associated with attention-deficit hyperactivity disorder [54]. The structural particularities of the different heteromers, for instance the interacting interfaces, underlie differential signaling [55,56,57]. In the case of the D_2_ short isoform, it so happens that it may interact with some of the D_4_ receptor variants but not with the D_4_._7_ isoform [58]. Whether this lack of homotropic DA receptor–receptor interaction has an impact on the pathophysiology of attention-deficit hyperactivity disorder is, at present, unknown.

## 4. Dopamine, L-DOPA and Parkinson’s Disease

Both imbalances in dopamine neurotransmission and alterations of brain circuits where dopamine is a key factor are involved in a variety of neurological and neuropsychiatric diseases, from alcohol/drug addiction to schizophrenia [18,19,20,21,22,23,24,25,59,60,61,62]. In fact, typical antipsychotics act “almost exclusively on the dopamine system” [63]. A review taking into account all those diseases is out of the scope of the present review, which will focus on the main dopamine-associated disease, namely, Parkinson’s disease (PD).

Details of DA synthesis are found in Figure 1. In the human, DA synthesis requires an essential amino acid, phenylalanine, which is the precursor of another relevant amino acid, tyrosine, that is the precursor of several bioactive molecules. Two enzymes are required for DA synthesis: L-tyrosine hydroxylase, which is a used as a marker of DA-producing cells/neurons, and L-3,4 dihydroxyphenylalanine (L-DOPA) decarboxylase. In “dopaminergic” cells, DA is the final product that can be re-used or degraded; however, in cells containing dopamine β-hydroxylase, e.g., cells in the adrenal gland, it is converted into noradrenaline. It has been considered that phenylalanine is essential because it would be quite “expensive” to synthesize it in neurons. Evolution has allowed mammals to obtain phenylalanine from nutrients for the easy procurement of tyrosine and its derivatives (DA, thyroxine, adrenaline, tyramine, etc.). This selective advantage has allowed us to save energy and synthetic enzymes. However, the hydroxylation of tyrosine to lead to L-DOPA requires reducing power, meaning that sustained DA production may lead to oxidative stress. To minimize the synthesis of new DA molecules, neurons have specific uptake mechanisms to incorporate interstitial DA and reuse it.

Parkinson’s disease (PD), resulting from the death of dopaminergic neurons of the substantia nigra, is characterized by motor symptoms, including tremors at rest and difficulty in performing movements. Consistent with the key role of DA in reward circuits, PD associates with impulsive control disorders. Some centers report that up to 40% of patients have impulsive tendencies, with hypersexuality being the most common, and men being most at risk (see Ref. [64] for review).

An efficacious drug still used today was suggested decades ago by the seminal work of Hornykiewicz and colleagues. After discovering that the cause of motor symptoms was a lack of DA in certain areas of the brain and noticing that DA is unable to cross the blood–brain barrier, they thought of using L-DOPA, i.e., its precursor. L-DOPA readily crosses the blood–brain barrier and is converted into DA in the brain. Fluctuations in the blood/brain levels of the drug and the need for a chronic treatment may lead to some side effects, mainly dyskinesia [65,66].

## 5. Dopamine Derivatives in Neurological and Neuropsychiatric Diseases

The route for DA degradation consists of two enzymatic steps catalyzed by monoamine oxidase and catechol-O-methyl transferase (COMT), which lead to homovanillic acid (IUPAC name: 2-(4-hydroxy-3-methoxyphenyl)acetic acid), a compound identifiable in the urine of healthy individuals (Figure 1). Obviously, the level in the urine decreases in untreated Parkinsonian patients, while it increases in patients receiving L-DOPA therapy. Homovanillic acid in the cerebrospinal fluid (CSF) is a surrogate marker for the catabolism of DA in the central nervous system [67]. The relevance of DA in the circuits involved in almost any higher function may be deduced from alterations in the levels of homovanillic acid in the CSF of patients with neurological diseases. An example is provided by tests in the CSF of 1388 children who were prescribed a lumbar puncture due to neurological clinical manifestations. The majority of patients had conditions that were not related to inheritable diseases related to DA. Among them, 696 had a clinical history that allowed stratification into the following categories, among others: epilepsy/epileptic encephalopathy (*n* = 206), dysmorphic traits/genetic syndromes (*n* = 69), motor disturbances (*n* = 65), hemorrhagic/ischemic injuries (*n* = 45) and mitochondrial disorders (*n* = 47). The level of the compound was significantly altered in the group of hemorrhagic or hypoxic/ischemic injuries. In addition, in patients with definitive diagnosis high HVA levels were seldom found, while low levels were mainly detected in infectious diseases of the CNS and in perinatal stroke [68]. In summary, measuring homovanillic acid in the CSF may be complementary but not essential for the diagnosis of neurological alterations [67,69].

In relatively mild conditions, DA may be spontaneously oxidized using molecular oxygen to produce DA quinones and free radical products. This possibility has attracted attention as the non-enzymatic degradation of DA leads to aminochrome, the precursor of the pigment that gives the particular dark color to the substantia nigra, neuromelanin. Several years ago, the oxidation of DA was linked to an excess of aminochrome and oxidative stress that could alter mitochondrial function and even produce autophagy in dopaminergic neurons (see Ref. [70] for review). A recent review has also addressed this issue [71].

The enzymes involved in the production and the degradation of DA (Figure 1) are very important in maintaining appropriate catecholamine levels and homeostatic conditions. Drugs that alter the activity of COMT, monoamine oxidase (MAO) or dopamine β-hydroxylase, may affect dopaminergic neurotransmission and any other DA-mediated physiological effects. However, the inhibition of COMT or MAO-B at adequate dosages may be beneficial for neuroprotection in PD [72,73]. The imbalance of systems that depend on DA may also occur following alterations in the expression of those enzymes, or polymorphisms in their respective genes [74,75].

## 6. Dopamine in the Gastrointestinal Tract

The notion of a gut–brain link is gaining momentum due to much evidence regarding the relevance of the composition of the gut microbiota to staying healthy and/or affecting the development or course of a disease. The communication starts by the regulation of the microbiota composition and intestinal function by molecules produced by either microbiota or gut cells. However, the long-distance regulations are mediated by cells of the immune system (see Refs. [76,77,78] for review). Although it is not completely canonical because the endocrine system involves the blood as a mediator of hormonal action, the concept of “microbial endocrinology” has been coined to delve into the symbiotic mechanisms that are established between our cells and the microbes that accompany us in our body [79].

As an example of the gut–brain link, there is cumulative evidence of the role of gut microbiota in the development of Parkinson’s disease. Although the underlying mechanisms are not known (see Ref. [80] for a review), it is reasonable to assume that the DA in the gut is key to a variety of physiological processes that may directly or indirectly impact on neurotransmission and neuronal fate in the CNS.

The amount of DA in the periphery is much higher than that in the CNS. More than 50 years ago, the release of DA by the rabbit ileum, likely coming from sympathetic nerve terminals, was demonstrated [81]. Apart from being produced by sympathetic neurons, DA in the periphery may be produced by a variety of cells. More than 50 years ago, the “staining characteristics” of so-called “dopamine cells” of the duodenal mucosa of cow or pig were reported [82]. Previously, a method for identifying monoamines in cells was reported by Falck [83]. DA in the gastrointestinal tract is surely produced by the phenylalanine coming from food digestion. The extra amino acid that is not processed in the intestinal cells is distributed, mainly through the blood, to all the organs of the mammalian body. In other words, not all phenylalanine coming from food digestion reaches the blood stream. On the one hand, microorganisms in the gut may both synthesize (using phenylalanine) or degrade neurotransmitters, DA included [84]. On the other hand, cells of the intestine or nerves in the intestine may accumulate and/or release DA. An example would be Paneth cells, whose cytoplasm contains aggregates of DA and 1-3,4-dihydroxyphenylalanine [85]. These are specialized cells, which are monoamine-secreting epithelial cells of the intestine located in the crypts of Lieberkühn. Paneth cells are involved in controlling the composition of the intestinal flora [86,87] and, remarkably, these cells together with enterocytes are in the front line of defense, thus having an intimate relationship with cells of the immune system [88,89].

A detailed description of all the possibilities of cells involved in phenylalanine and DA processing is out of the scope of the present review. However, it is worth assuming that DA locally produced is impacting cells of the gastrointestinal tract, and cells in Peyer’s patches and mesenteric lymph nodes (Figure 2). Non-neuronal cells surely respond to DA, but further research is needed in order to know how to take advantage of gut DA and DA-rich gastrointestinal cells to combat PD and inflammatory bowel diseases [90]. Immune cells, which express different DA receptor types, play a key role (see next section). A prominent role of polymorphisms of the DRD2 gene, which codes for the D2 receptor, has been associated with the risk and the course of inflammatory bowel disease and the efficacy of the treatment; the authors of the study suggested that the receptor might be a target for Crohn’s patients refractory to the medication [91].

## 7. Dopamine, Immune Cells, Inflammation, Autoimmunity and Parkinson’s Disease

Being produced by neurons, or not, DA acts in a wide variety of cells that express DA receptors. Generally speaking, DA is not an endocrine hormone, i.e., it does not travel by blood to reach cells in different places in the mammalian body. DA is more a molecule that acts locally; analogous to synaptic action, the compound may act in different set ups where cells are exchanging information. Therefore, for DA to act in the gut and have an impact in the CNS (and other systems), intermediate cells are needed. One of the best examples of the local action of DA occurs at the so-called immunological synapse where dendritic cells interact with lymphocytes. Additionally, DA is present in lymph nodes, and not only in those of the gastrointestinal tract. Although the role of another “neurotransmitter”, glutamate, has been more studied in the immunology research field [92], DA plays a key role in the germ center, where B and T lymphocytes interact. A subpopulation of T helper cells releases DA that “accelerates productive synapses in germinal centers” [93].

DA acting on T-cells may contribute to neurodegeneration in PD. On the one hand, the expression of the mRNA transcripts of D_3_ receptors is altered in blood lymphocytes from patients [94]. On the other hand, the deletion of the D_3_ receptor gene or the pharmacological blockade using receptor antagonists in rodent models of PD protect against neuroinflammation and dopaminergic denervation [95,96]. It is tempting to speculate that DA occupies a central place in the gastrointestinal manifestations of PD patients, also constituting an important piece in the gut–brain communication puzzle occurring in PD [97,98,99,100]. Although DA acting on D_3_ receptors is noxious in PD models, the deletion of the receptor leads (in animal models) to chronic depression and anxiety [101].

Via the regulation of cAMP levels and mitogen-activated protein kinase (MAPK) pathway activation, D_3_ but also D_5_ receptors regulate T lymphocyte activation and contribute to the immune response in a variety of diseases [102]. For instance, D_3_ receptor-mediated actions in CD4^+^ cells favor Th1/Th17-mediated immunity, thus favoring the inflammatory potential of the lymphocytes [103]. The activation of D_5_ receptors in myeloid antigen-presenting cells facilitates the development of an experimental model of autoimmune disease of the nervous system (encephalitis) [104]. Overall, DA plays an important role in the immune system, regulating the functionality of dendritic cells and lymphocytes and, in addition, there is evidence of its important role in the development of autoimmune diseases [105,106]. The often-found detrimental action exerted by DA acting on lymphocytes opens up therapeutic opportunities. Thus, the inhibition of DA receptor-mediated actions may be beneficial in autoimmune disorders. In addition, it has been proven in an experimental model that the blockade of D3 receptor-mediated signaling in dendritic cells favors T-cell-mediated anti-tumor activity [107].

## 8. Future Perspectives of DA as Neurotransmitter

DA is associated with more diseases than Parkinson’s, and exerts actions that go beyond the substantia nigra and the striatum. Dopamine receptors are in the reward circuits in the brain, and are related, for instance, to plastic changes occurring in drug addiction. As above-mentioned, DA receptors are extrasynaptically located and participate in, among other things, brain remodeling/plasticity. Accordingly, future research must address (i) DA receptor (including heteromers) distribution in different areas of the brain in both health and disease, and (ii) the circuitry associated with executive actions (both in health and disease).

Knowledge of the reasons why gambling behavior is more widespread in Parkinsonian men than in Parkinsonian women is lacking. This is an example of what is needed to know and can be figured out by ad hoc research using animal models or in the human brain (e.g., by in vivo positron emission tomography or by immunochemical methods on post mortem samples). Addressing this topic can not only serve to help better manage the disease, but can also provide clues as to what may be different in the human brain according to gender: are the sex hormones that give this differential trait in the circuits related to PD? This is one of the many interesting questions that arise when considering DA as a neurotransmitter of the CNS in its broad sense: both in wiring and in volume neurotransmission.

## 9. Future Perspectives of DA as Regulatory Molecule

The quite recently discovered link that, via dopamine, is established between the gut and the CNS is very attractive in terms of understanding both how our body works and the pathophysiological features of some diseases. In fact, our body is constituted by our cells, but also by exogenous cells, such as those constituting our microbiota. What is the DA production by microbiota in a healthy gut? Does this DA contribute to that keep pathogens under control by impacting DA receptors in the immune cells within the gastrointestinal tract? Or, otherwise, does the excess or defectiveness of DA in the intestine disrupt such homeostatic control?

When PD appears and there is evidence of a “dopamine link” between the gut and the CNS, are events in the gut key for the development of the disease? One conservative hypothesis would be that PD starts in the brain, and that the gut–brain link may serve to impact on the denervation rate. Depending on the content of the microbiota and the status of the immune system components in the gut, primed immune cells (Figure 2) may reach the brain via infiltrating lymphocytes to boost neuroinflammation and the death of dopaminergic cells in the substantia nigra. The hypothesis may be wrong, but addressing it with creative research approaches will surely help to better understand how different systems interact in both healthy and pathological situations.

There are several questions that can be raised in this exciting field of long-distance interactions between different organs, with the participation of exogenous cells and having DA as a mediator.

## Figures and Tables

**Figure 1 biomedicines-09-00109-f001:**
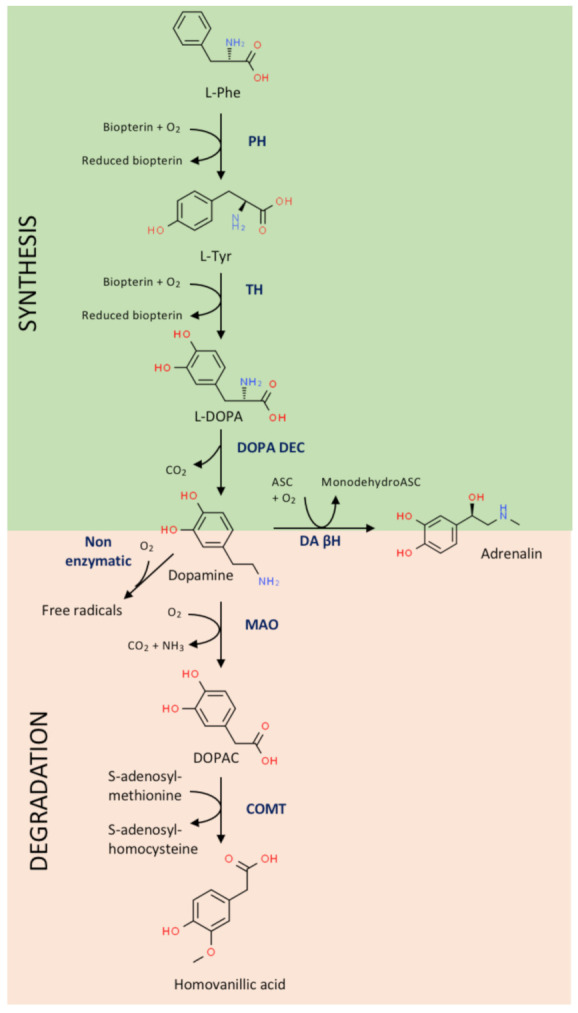
Dopamine synthesis and degradation pathways. ASC, ascorbic acid; COMT, catechol-o-methyltransferase; DA βH, dopamine β-hydroxylase; DOPAC, 3,4-dihydroxyphenylacetic acid; DOPA DEC, L-DOPA decarboxylase; L-DOPA, levo-dopa; L-Phe, L-phenylalanine; L-Tyr, L-tyrosine; MAO, monoamine oxidase; PH, phenylalanine hydroxylase; TH, tyrosine hydroxylase.

**Figure 2 biomedicines-09-00109-f002:**
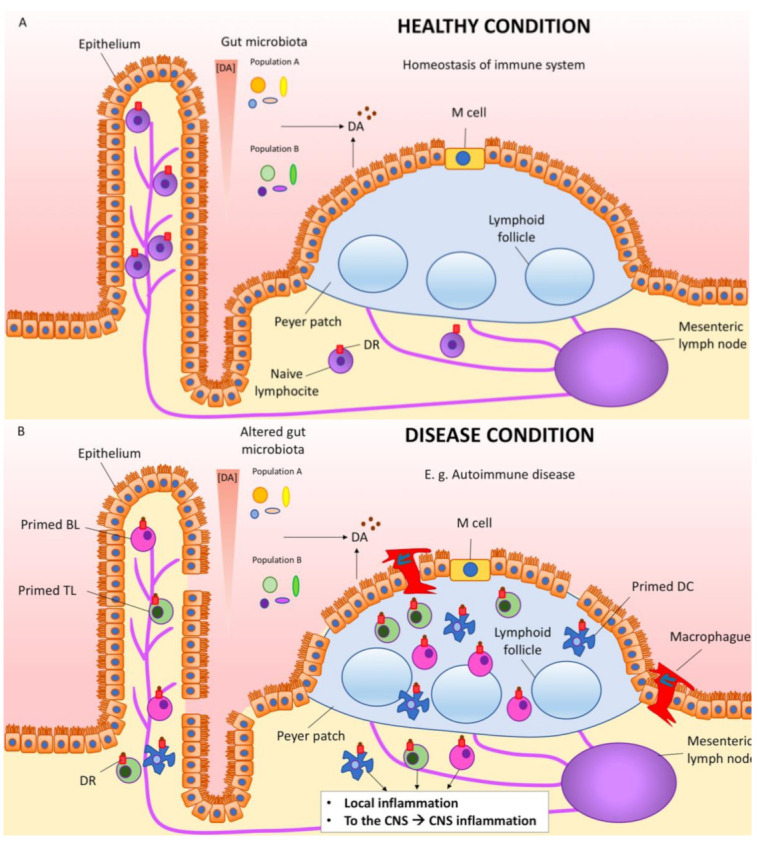
Dopamine link between gut and inflammation in the periphery and in the CNS. Cells in the intestine and also gut microbiota cells produce DA, which on one hand influences the composition of the gut microbiota and on the other hand binds to the dopamine receptors located in the surrounding cells, including immune cells. (**A**) In healthy conditions, there is homeostasis of the gut immune system. (**B**) In diseases affecting the gut, e.g., autoimmune diseases such as inflammatory bowel disease, immune cells (DCs, TLs, BLs, etc.) get primed. Macrophages infiltrate from the intestinal lumen trough the damaged epithelium. These primed immune cells produce local inflammation, but also reach the blood and may permeate the blood–brain barrier, reaching the brain and boosting CNS inflammation. BL, B lymphocyte; DC, dendritic cell; DA, dopamine; DR, dopamine receptor; TL, T lymphocyte.

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
