# Peer review of "Dopamine in Health and Disease: Much More Than a Neurotransmitter"

_biomedicines, 2021, doi:10.3390/biomedicines9020109_

Round 1
Reviewer 1 Report
This review describes dopamine in health and disease, and its receptors, its role in the CNS and neuropsychiatric disorders but also the connection between gut and CNS, associating dopamine receptors in different cells of the immune system where dopamine regulates various processes such as antigen presentation, T-cell activation, and inflammation.
It is nicely written. However, besides DA receptors, I miss the data on the alterations in COMT, DBH and MAO. While the authors mention Parkinson’s disease, they did not mention any other dopamine related neuropsychiatric diseases; they mentioned DRD2 polymorphisms related to inflammatory bowel disease, but on the other hand, they did not describe any of the dopamine system-related polymorphisms associated with ADHD, drug or alcohol abuse etc; I missed some description of these disorders and genetic variants associated with them.
Page 6, lines 170-207 and page 6/7, lines 208-236 the same paragraph is written „5. Dopamine derivatives in neurological and neuropsychiatric diseases.“ And 6. Dopamine derivatives in neurological and neuropsychiatric diseases. Therefore, please delete one paragraph.
Author Response
This review describes dopamine in health and disease, and its receptors, its role in the CNS and neuropsychiatric disorders but also the connection between gut and CNS, associating dopamine receptors in different cells of the immune system where dopamine regulates various processes such as antigen presentation, T-cell activation, and inflammation.
Answer: Thanks for the nice comments on the paper.
It is nicely written. However, besides DA receptors, I miss the data on the alterations in COMT, DBH and MAO. While the authors mention Parkinson’s disease, they did not mention any other dopamine related neuropsychiatric diseases; they mentioned DRD2 polymorphisms related to inflammatory bowel disease, but on the other hand, they did not describe any of the dopamine system-related polymorphisms associated with ADHD, drug or alcohol abuse etc; I missed some description of these disorders and genetic variants associated with them.
Answer: We appreciate these comments that have been taken into account in the revised version of the paper. We already commented D4R polymorphisms associated to ADHD:
“The circa 20 variants of the D4 receptor are due to a hypervariable region of the D4 gene, they come from 27 haplotypes [44,45] and one of them, the D4.7, is associated to the attention-deficit hyperactivity disorder [46]”
In the revised version we have added refs that link dopamine receptor polymorphisms to drug and alcohol abuse.
Page 6, lines 170-207 and page 6/7, lines 208-236 the same paragraph is written „5. Dopamine derivatives in neurological and neuropsychiatric diseases.“ And 6. Dopamine derivatives in neurological and neuropsychiatric diseases. Therefore, please delete one paragraph.
Answer. Thanks for noticing. The paragraph has been removed
Minor changes are not highlighted; other modifications are highlighted in yellow
Reviewer 2 Report
This sufficiently detailed, up-to-date, and generally well written review on an important subject almost completely ignores the microbiological aspects of the dopamine function, excep fr one picture, the statement on p. 7 of manuscript that "microorganisms in the gut may both synthesize (usingnphenylalanine) or degrade neurotransmitters, DA included" and the question in the Conclusion section " What is the DA production by microflora in a healthy gut?" The authors should reference relevant recent works in which much attention is paid to the microbiota's role in terms of both producing dopamine and related neurochemicals and specifically responding to them. The manuscript does not provide information on the pathogenic microorganisms-stimulating effect of DA and other catecholamines. This is of primary clinical relevance, since an infected organisms produces large amounts of DA and other catecholamines in terms of its stress response, and their stimulation of the pathogen's growth and virulence causes a dangerous vicius circle. I'm convincd that at least the following recent publications (in which this issue is discussed in detail) should be added to the otherwise somewhat too short reference list:
Lyte, M. (2016). Microbial endocrinology: an ongoing personal journey. In: M. Lyte (Ed.), Microbial Endocrinology: Interkingdom Signaling in Infectious Disease and Health (pp. 1-24). New York: Springer.
Oleskin, A. V., El’-Registan, G. I.,& Shenderov, B. A. (2016). Role of neuromediators in the functioning of the human microbiota: “business talks” among microorganisms and the microbiota-host dialogue. Microbiology (Russia), 85(1), 1-22.
Oleskin,A. V., Shenderov, B. A., & Rogovsky,V. S. (2017). Role of neurochemicals in the interaction between the microbiota and the immune and the nervous system of the host organism. Probiotics & Antimicrobial Proteins, 9(3), 215-234.
Oleskin, A. V., & Shenderov, B. A. (2019). Probiotics and psychobiotics: the role of microbial neurochemicals. Probiotics & Antimicrobial Proteins, 11(4),1071-1085
Oleskin, A. V. & Shenderov, B. A. (2020). MICROBIAL COMMUNICATION AND MICROBIOTA-HOST INTERACTIONS: BIOMEDICAL, BIOTECHNOLOGICAL, AND BIOPOLITICAL IMPLICATIONS. Nova Science Publishers, Inc, New York, United States
There are a large number of new experimental data chracterizing DA as a universal biological signal molecule. For instance it is produced by plants including those used as food or spices (Boyang,C., Oleskin, A. V., & Tatiana, V. (2020). Detecting biogenic amines in food and drug plants with HPLC:Medical and nutritional implications. Journal of Pharmacy and Nutrition Sciences, 10(3),88–91); DA is present in fermented tea (Shanenko, E. F., Efremenkova, O. V., Mukhamedzanova, T. G., & et al.. (2020). Using the eurotium cristatum fungus for preparing fermented herbal teas. Journal of Pharmacy and Nutrition Sciences, 10,341–361) and this is how it can reach the intestines.
There are some minor terminological problems in the manuscript. The authors interchangeably use "(nor)adrenalin" and "(nor)adrenaline)", while only the latter is predominantly prefererred in the literature. The term "microflora" is utdated and should be consistently replaced with "microbiota". The language is not without problems either; e.g., some articles are missing, exemplified by the phrase "and signal is transmitted" on p.2, line 76-77. I believe, therefore, that a minor revision of the text is necessary.
Author Response
his sufficiently detailed, up-to-date, and generally well written review on an important subject almost completely ignores the microbiological aspects of the dopamine function, excep fr one picture, the statement on p. 7 of manuscript that "microorganisms in the gut may both synthesize (usingnphenylalanine) or degrade neurotransmitters, DA included" and the question in the Conclusion section " What is the DA production by microflora in a healthy gut?" The authors should reference relevant recent works in which much attention is paid to the microbiota's role in terms of both producing dopamine and related neurochemicals and specifically responding to them. The manuscript does not provide information on the pathogenic microorganisms-stimulating effect of DA and other catecholamines. This is of primary clinical relevance, since an infected organisms produces large amounts of DA and other catecholamines in terms of its stress response, and their stimulation of the pathogen's growth and virulence causes a dangerous vicius circle. I'm convincd that at least the following recent publications (in which this issue is discussed in detail) should be added to the otherwise somewhat too short reference list:
Lyte, M. (2016). Microbial endocrinology: an ongoing personal journey. In: M. Lyte (Ed.), Microbial Endocrinology: Interkingdom Signaling in Infectious Disease and Health (pp. 1-24). New York: Springer.
Oleskin, A. V., El’-Registan, G. I.,& Shenderov, B. A. (2016). Role of neuromediators in the functioning of the human microbiota: “business talks” among microorganisms and the microbiota-host dialogue. Microbiology (Russia), 85(1), 1-22.
Oleskin,A. V., Shenderov, B. A., & Rogovsky,V. S. (2017). Role of neurochemicals in the interaction between the microbiota and the immune and the nervous system of the host organism. Probiotics & Antimicrobial Proteins, 9(3), 215-234.
Oleskin, A. V., & Shenderov, B. A. (2019). Probiotics and psychobiotics: the role of microbial neurochemicals. Probiotics & Antimicrobial Proteins, 11(4),1071-1085
Oleskin, A. V. & Shenderov, B. A. (2020). microbial communication and microbiota-host interactions: biomedical, biotechnological, and biopolitical implications. Nova Science Publishers, Inc, New York, United States
Answer: We appreciate the comment. This issue is, at present, poorly characterized. We have expanded this issue in the revised version also incorporating those references. This issue is still in initial phases of research, i.e. there is a lot of research to do and with potentially relevant findings in the future. We wonder about 86 being considered a “somewhat too short” list of references. The last reference (Oleskin, A. V. & Shenderov, B. A. (2020) is not found in databases.
There are a large number of new experimental data chracterizing DA as a universal biological signal molecule. For instance it is produced by plants including those used as food or spices (Boyang,C., Oleskin, A. V., & Tatiana, V. (2020). Detecting biogenic amines in food and drug plants with HPLC:Medical and nutritional implications. Journal of Pharmacy and Nutrition Sciences, 10(3),88–91); DA is present in fermented tea (Shanenko, E. F., Efremenkova, O. V., Mukhamedzanova, T. G., & et al.. (2020). Using the eurotium cristatum fungus for preparing fermented herbal teas. Journal of Pharmacy and Nutrition Sciences, 10,341–361) and this is how it can reach the intestines.
Answer. These are very important pieces of information but, honestly, in our opinion they do not fit into our paper. We focus in mammals/humans. We would like to point out that few molecules taken orally (e.g. in herbal teas) by humans reach the intestine without being transported into the blood or metabolized. Also, as far as we understand, the dopamine in plasma is not “relevant”. As we indicated in the first version, dopamine multiple effects do not seem to be endocrine.
There are some minor terminological problems in the manuscript. The authors interchangeably use "(nor)adrenalin" and "(nor)adrenaline)", while only the latter is predominantly prefererred in the literature. The term "microflora" is utdated and should be consistently replaced with "microbiota". The language is not without problems either; e.g., some articles are missing, exemplified by the phrase "and signal is transmitted" on p.2, line 76-77. I believe, therefore, that a minor revision of the text is necessary.
Answer. Thanks for these comments that have been considered on preparing the revised version. The paper has been carefully checked before resubmission. It must be highlighted that in my computer the IJMS template prevent use of the dictionary. Something strange but once the words/figs are into the IJMS template I cannot perform automatic corrections.
Minor changes are not highlighted; other modifications are highlighted in yellow